# Analysis of the Spatial and Temporal Evolution of Urban Resilience in Four Southern Regions of Xinjiang

**DOI:** 10.3390/ijerph20065106

**Published:** 2023-03-14

**Authors:** Ying Li, Yonggang Ma, Junjie Liu, Jianjun Yang

**Affiliations:** 1School of Geography and Remote Sensing Science, Xinjiang University, Urumqi 830049, China; 2Xinjiang Key Laboratory of Oasis Ecology, Xinjiang University, Urumqi 830046, China; 3School of Ecology and Environment, Xinjiang University, Urumqi 830049, China; 4Xinjiang Jinghe Observation and Research Station of Temperate Desert Ecosystem, Ministry of Education, Xinjiang 830046, China

**Keywords:** urban resilience, safe development, four regions of southern Xinjiang, scale-density-morphology

## Abstract

Resilience theory plays a pivotal role in promoting sustainable urban development and the long-term stable development of the national economy. Based on the “scale-density-form” model of urban resilience, this paper shifts the perspective of urban resilience from the eastern region with higher level of economic development and better infrastructure construction to the arid northwest region with more fragile ecological environment and weaker urban development potential, which enriches the connotation and mechanism of urban resilience to a certain extent. Using ArcGIS platforms, statistical data and remote sensing data as data sources, this paper analyzes the urban resilience of four southern Xinjiang regions (Aksu Administrative Office, Kashgar Administrative Office, Kizilsu Kirgiz Autonomous Prefecture and Hotan Administrative Office) in time and space from 2000 to 2020 using a three-dimensional resilience analysis framework based on scale, density and morphology. The results show that the urban de-development of the study area faces a strong scale safety constraint due to the small available land area in the study area, which leads to its small urban construction land area. The county- and city-scale elasticity levels of Aksu Administrative Office and Kashgar Administrative Office are higher than the average level of the study area, while most of the counties and cities in Kizilsu Kirgiz Autonomous Prefecture and Hotan Administrative Office are lower than the average level of the study area, with large differences between counties and cities. The geographical location of the study area determines the backwardness of the region in terms of ideology, production methods and technology, which seriously restricts the development of local society and economy. In terms of density resilience, there are large differences among counties and cities in the study area, and the density resilience of Aksu, Kashgar and Kucha is much higher than that of other counties and cities. In terms of morphological resilience, with the increasing prominence of ecological status, the urban landscape layout of the study area has changed significantly, leading to changes in the distance between its blue-green landscape and gray-white landscape, which in turn causes changes in morphological resilience. Based on the above findings, initiatives and paths for resilience regulation in the study area are proposed in terms of scale, density and morphology. The study also has a reference value for local urban safety development.

## 1. Introduction

The word resilience is often used to describe the flexibility or stretching of matter [1]. Alexander analyzed the word resilience from an etymological point of view, as the word is derived from the Latin word “resilio” meaning “to jump back” and later was widely defined as meaning “to return to the original state” [2,3]. Resilience was first introduced as an academic concept in the field of physics to describe the ability of a material to absorb external impact forces [4]. In 1818, Tredgold used the term resilience to explain the ability of different types of wood to adapt to different loads without breaking, thus describing the properties of wood [5]. In the 19th century, with the development of the Industrial Revolution in the West, resilience was used in physics and mechanics to describe the stability of materials and their ability to resist external disturbances; in the 1950s and 1980s, Western psychological research used the term resilience in the field of pathogenesis and psychopathology to describe the ability to recover after mental trauma [6]. In 1996, Holling expanded resilience to the field of ecological resilience [7]. Since the 1990s, due to its acceptance by economists and geographers, the concept of resilience began to transition to anthropology, disaster science, economics, sociology, urban and rural planning and other social sciences, and has been rapidly promoted and developed, producing a large number of relevant research results.

The term urban resilience has not been clearly defined, and the existing concept of resilient cities does not perfectly integrate the key concepts of resilience theory and urban theory, and is still under debate [8,9,10,11,12,13,14], while a new definition of urban resilience has been proposed, i.e., “urban resilience refers to the capacity of an urban system and all its components across spatial and temporal scales of socio-ecological and socio-technical networks to maintain or rapidly recover the required functions and adapt to change in the face of disruptions, and to rapidly transform systems that limit their adaptive capacity today or in the future”. As a theoretical framework of urban resilience used in research, Bruneau proposed the “TOSE” framework to further enrich the meaning of urban resilience, pointing out that the elements of urban resilience should include technological resilience, organizational resilience, social resilience and economic resilience [15]. Cutter proposed the disaster resilience of place (DROP) framework with the goal of disaster management, arguing that urban resilience is a continuous process that overemphasizes a single disaster response process [16]. Nyström proposed a framework for the sustainability of coupled human–environment systems (the resilience of coupled human–environment systems, HES) by linking human–environment systems and resilience research [17]. As a bridge between theory and practice, urban resilience assessment not only recognizes the impact of random disturbances and limited capacity on cities, but also emphasizes the integrity of the urban pattern and the continuity of its function. Urban resilience has become an important evaluation unit for safe urban development, and the current research focuses on methods of evaluating urban resilience and systems of evaluation indices [18,19,20,21,22,23,24,25,26].

The urbanization process of the past 200 years has promoted the progress of human civilization; at the same time, due to a lack of comprehensive governance concepts and ideas about development, environmental pollution and ecological damage have brought various shocks and risks to cities, and these have continuously increased [27]. In this context, a series of comprehensive views on urban conditions and assessing them from a new perspective, represented by urban resilience, has attracted the wide attention of researchers and policy makers, and resilient cities, alongside habitable cities and sponge cities, have become mainstream ideas in urban development and construction [28].

Most of the existing urban resilience studies are focused on the eastern region, and fewer cities have formulated resilience strategies and action plans and incorporated the concept of resilience into the construction and daily management of cities. The level of economic development and urbanization in Xinjiang is relatively low, and the ecological environment is relatively fragile. The counties and cities in the four southern prefectures of Xinjiang are mainly located in the marginal areas of the Tarim Basin, and the natural environment varies greatly among the counties and cities, with special geomorphological conditions. The development of economy brings many adverse effects to humanities, society and ecology, and the economic and ecological development is not coordinated, the gap between urban and rural areas is large and the sensitivity and vulnerability is very obvious, thus affecting urban security, which needs urgent attention. Therefore, the study of urban resilience in poor border areas, taking the four southern Xinjiang regions as an example, can provide basic data for planning in the context of territorial spatial planning and help decision makers to formulate resilience action plans to enhance urban resilience and coordinate sustainable urban and rural development. It is also useful for decision makers to formulate resilience action plans to enhance urban resilience and integrate sustainable urban and rural development. 

## 2. Materials and Methods

### 2.1. Research Area

The four southern Xinjiang prefectures are located in the periphery of the Tarim Basin south of the Tianshan Mountains and north of the Kunlun Mountains in Xinjiang, China, at longitudes 73°26′ E–84°07′ E and latitudes 35°28′ N–41°54′ N. the region mainly includes four cities and towns, namely, Kashgar Administrative Offices, Hotan Administrative Offices, Aksu Administrative Offices and Kizilsu Kirgiz Autonomous Prefecture, with a total area of about 613,600 km^2^, accounting for about 36.96% of the total area of Xinjiang. The population is approximately 10.34 million, accounting for 39.94% of the total population of Xinjiang. The climate of the four southern Xinjiang regions is extremely arid; the Gobi Desert is widespread, the ecological environment is very fragile, and sandstorms, earthquakes, drought, high winds, hail and other natural disasters are widely distributed, and have the characteristics of high frequency, intensity and disaster losses. In recent years, with the implementation of the national strategy of “One Belt and One Road”, the counterpart assistance to Xinjiang and the new urbanization development strategy of Xinjiang, the four southern Xinjiang regions have made significant developments in many aspects such as ecological civilization construction, economic and social development and urban infrastructure construction. However, due to the fragile ecological environment and backward production methods, the economic development of the four southern Xinjiang regions still lags behind the average level of the autonomous region, with very difficult production and living conditions, backward infrastructure construction and slow socioeconomic development. There are rich border tourism resources within the region, as shown in Figure 1.

### 2.2. Data Sources

The data required in this paper mainly include spatial data and socioeconomic data. Among the spatial data, DEM data are from the Geospatial Data Cloud, NDVI data are from the official NASA website, population density data are from the World Population Dataset, road network data are from the Open Street Map, land use data are from the Resource and Environment Data Center of the Chinese Academy of Sciences, and ecological protection red line data are provided by the subject group. This paper used GIS software to resample the spatial data into 30 m × 30 m grids for normalization to maintain the uniformity of the data, and then conducted spatial analysis of each dataset and its construction land suitability evaluation. The socioeconomic data were mainly obtained from the Xinjiang Statistical Yearbook and the County Statistical Yearbook (2001–2021). Individual missing data in the socioeconomic data were expressed as the average of adjacent years to indicate their values in that year. The data sources are shown in Table 1.

### 2.3. Measures of Urban Resilience

#### 2.3.1. Meaning of Scale-Density-Form in Urban Resilience

The integration of urban resilience theory and practice is the basic strategy to realize “city building, planning first”. According to the connotation of resilient cities, this paper believes that resilient cities should have two conditions: first, to reduce the occurrence of disasters and achieve rapid urban recovery after disasters, which is mainly controlled by territorial planning; and second, to achieve good and orderly development of the ecological environment in urban expansion. In other words, the expansion of urban development boundaries should be controlled and the ecological environment should be protected.

The size of a city was expressed in terms of the total urban population and urban land, and the population size was usually used as the decisive indicator to evaluate its size. Currently, the city scale directly affects the city strategic positioning, urban master plan and other urban development and urban layout, which has attracted more extensive attention from planning scholars [29]. The biggest manifestation of urban scale in urban safety is the creation of “big city disease” and the resulting urban management mistakes [30]. From the perspective of territorial planning, urban scale resilience was based on urban spatial expansion and represents the limit of urban scale development and maximum capacity [31]. To a certain extent, urban scale security constrains the scope and speed of urban expansion. For a long time, the focus on urban development has mostly emphasized high economic growth, leading to the blind expansion of urban scale, resulting in a serious waste of land resources, environmental damage and other problems [32]. Based on this, the theories of compact cities and urban growth boundaries have also been proposed to alleviate urban sprawl to a certain extent in order to effectively optimize urban spatial forms [33].

Urban density resilience was the load condition of human activities in urban space form, and density security is the precondition for sustainable urban development. High-density urban development can, to a certain extent, guarantee the relatively low cost of urban construction and the effective transition of urban economic structure transformation and upgrading by increasing the supply in space [34]. Therefore, we usually use urban density as a criterion to formulate the strategy of urban infrastructure and the intensity of planned development. In the West, quantifiable indicators such as “residential density (buildings/site)”, “residential density (residents/site)”, and “job density (jobs/site)” are usually used to measure urban density [35,36,37]. In China, floor area ratio (FAR), building density and green space ratio are used to control the development intensity of land parcels [38,39,40]. For a long time, discussions on urban density have focused on linking urban form to urban living. To some extent, the control of urban density is mainly to ensure people’s living needs. As a frontier region, urban development in the four southern border states still needs to focus more on improving economic development, continuously improving and building urban infrastructure and other aspects to improve the wellbeing of residents and promote high-quality urban development.

Urban morphology refers to the physical–spatial layout of the city as well as the development pattern [41]. It encompasses the spatial configuration and organization of ecological environment, planning layout, history and culture, and other elements [42]. The ecological environment is the background requirement for urban spatial development and is one of the intrinsic factors for the creation of urban morphology [43]. While scale and density have the most prominent impact on urban development when cities face serious unexpected disasters or long-term pressures on development existence, urban morphology also plays an auxiliary role in mitigating or accelerating this process. Only the simultaneous expansion of ecological land along with the continuous expansion of urban construction land can effectively improve the urban environment and dissipate the adverse effects of large built-up areas, thus enhancing urban resilience. At the same time, the decentralized layout of urban construction land can also improve the self-organization and relative independence of cities, thus increasing urban resilience.

Based on the connotation of scale-density-form urban resilience, this paper draws on the scale-density-form urban resilience evaluation framework proposed by previous authors and uses the polyhedral method to measure the comprehensive urban resilience, reflecting the joint influence of scale, density and form on urban resilience. The polyhedral approach is used to measure the comprehensive urban resilience, reflecting the joint influence of scale, density and form on urban resilience. The evaluation of the suitability of land for construction is used to delineate the maximum scope of urban expansion, to further limit the increase of urban construction area in 33 counties and cities in the four southern prefectures, and to compare it with the developed land for construction, so as to assess the urban scale resilience; the index weights of each evaluation index in economic and social aspects are calculated based on the entropy value method and the hierarchical analysis method, and then the final weights are obtained by combining the two methods. Based on the final weights, urban density resilience is evaluated; the land use types in the study area are classified into “source landscape” and “sink landscape” based on the 30m land use data of CAS, and then the “source–sink” landscape distance index, statistical analysis, and the “source–sink” landscape distance index are used. The “source–sink” landscape distance index, statistical analysis and other tools were used to calculate the urban form resilience index of each county and city in the study area. Finally, the comprehensive urban resilience index of each county and city in the study area is measured according to the polyhedron method in order to prepare for the spatial and temporal evolution and detection of influencing factors. A flow chart of the evaluation is shown in Figure 2.

#### 2.3.2. Analysis of Scale Resilience

A city’s resilience is based on urban spatial expansion and represents the limit and maximum capacity of city-scale development. The safety of a city in terms of scale restricts the scope and speed of urban expansion to a certain extent. For a long time, most of the focus on urban development has emphasized rapid economic growth, resulting in the blind expansion of the urban scale, leading to serious wastes of land resources, environmental damage and other problems. Based on this, the state has successively issued relevant policies, delineated areas of permanent basic farmland, defined ecological protection zones, etc., to control urban expansion and prevent excessive urban growth, and thus achieve high-quality urban development. Theories such as compact cities and urban growth boundaries can alleviate disorderly urban expansion and effectively optimize the urban spatial form. Based on an assessment of the suitability of land development, combined with the actual situation of the research area and related research results [44,45,46], seven indicators were selected to cover the aspects of natural conditions, socioeconomic factors and ecological factors to construct a system of evaluating the suitability of construction land in the study area, to obtain the maximum scope of urban expansion and compare it with the developed construction land to evaluate the resilience of the cities’ scale. The specific grading standards are shown in Table 2.

According to the fragile ecological environment and construction development needs of the four southern counties, the suitability of construction land in each county and city was evaluated using restrictive factors, and the suitable construction land was classified as high suitable, medium suitable, low suitable and unsuitable by using the overlay analysis of ArcGIS software. The results are shown in Figure 3.

Based on the evaluation results of construction land suitability obtained from Figure 2, the suitable construction land area of each county and city was extracted. Using the land use data obtained from the platform of the Resource and Environment Data Center of the Chinese Academy of Sciences, the construction land area of each county and city in the study area in 2000, 2005, 2010, 2015 and 2020 was obtained, and the scale resilience index of each county and city in the four southern Xinjiang prefectures was measured using the formula, considering the suitability of construction land and the current situation of land use in the study area.
(1)$$A_{S}=X_{S}/Y_{d}$$
where A_S_ is the scale resilience index; X_S_ is the area of suitable land for construction; and Y_d_ is the area of developed land for construction in five years.

#### 2.3.3. Analysis of Density Resilience

The density elasticity of a city refers to the loading conditions of the intensity of human urban activities. Security in terms of density is a prerequisite for sustainable and safe development. Urban density reflects the high density of human activities, and a highly dense population seems to bring problems such as resource shortages, pollution and deterioration of the living environment, which in turn affects the safe development of the city. Therefore, we usually use urban density as a criterion to formulate strategies and plan the intensity of urban infrastructure construction. As an economically underdeveloped region, the urban development of the four southern border states still focuses on improving the level of economic development and infrastructure construction; therefore, in this paper, based on the reference of relevant literature and considering the availability of the data [47], we characterize the resilience of urban density in terms of both economic development and public service facilities. The index system is shown in Table 3.

For each indicator in this system, it is necessary to determine the weights to distinguish the degree of importance of each indicator. The entropy value method and the hierarchical analysis method were used to calculate the indicators’ weights separately, and then the two methods were combined to obtain the final weights. According to the final weights, the urban density resilience of the four southern regions was calculated as follows.

(1)Entropy method

Entropy is a thermodynamic concept used to describe the degree of chaos of a system, which was first introduced into information theory by Shannon to characterize the uncertainty of the information source signal [48], called information entropy (entropy for short), and has been widely used in engineering technology, social economy and other fields. The entropy method can avoid the bias brought by human factors, reflect the relative intensity of each indicator in the competitive sense under the condition that the value of various evaluation indicators is determined, and can quantify some factors with unclear boundaries and those not easy to quantify [49]. The entropy method to determine the weights is judged by calculating the size of variability of each indicator. If the information entropy of an indicator is smaller, the greater its variability and the more information it covers, the greater its weight, and vice versa. The specific calculation method refers to the literature [50].

(2)Hierarchical analysis

Hierarchical analysis is a systematic method of urban density elasticity evaluation by decomposing urban density elasticity as a system into two aspects: economic density and public infrastructure, selecting 10 indicators as its evaluation factors, and calculating each indicator factor and the weights of its evaluation factors through the fuzzy quantification method of qualitative indicators. Its advantage is that it can deal with the combination of qualitative and quantitative issues, and can import and quantify the subjective judgment and policy experience of decision makers into the model. In order to simplify the calculation method, this paper uses software to calculate its indicator weights.

(3)Combination of entropy method and hierarchical analysis

Combination weights are new weights made by combining the weights calculated by the entropy value method and hierarchical analysis method. This calculation process combines the entropy value method and the hierarchical analysis method, which makes the weight calculation consider not only the objectivity of its selected factors, but also the subjectivity of the evaluation factors. Its calculation method is as follows.
(2)$$wj=\frac{w_{1j}*w_{2j}}{\sum_{j=1}^{n}w_{1j}*w_{2j}}$$
(3)$$A_{D}=\sum_{j=1}^{n}wj*Aij$$
where $w_{1j}$ is the weight calculated by the hierarchical analysis method; $w_{2j}$ is the weight calculated by the entropy value method; A_D_ is the density resilience index of the ith city; A_ij_ is the standardized value of the indicator; and W_j_ is the weight of the jth indicator.

#### 2.3.4. Analysis of Urban Morphological Resilience

Source–sink landscape theory is the basis for correlating urban spatial layout and ecological processes [51]. According to the “source–sink” theory, the heterogeneous landscape can be divided into “source landscape” and “sink landscape”. Source and sink landscapes represent the landscape types that promote and prevent ecological development processes, respectively [52]. A sink landscape can absorb and offset the negative effects brought by a source landscape. In terms of urban ecological security, the larger the area occupied by the blue and green landscape, the better, within a certain area. Its spatial layout is particularly important, such as the uneven configuration of the source and sink landscape, which can lead to an urban heat island effect. In this paper, using 30 m land use data from CAS as the data source, we take the gray landscape (urban construction land) as the source landscape and the blue-green landscape (woodland, grassland, water bodies) as the sink landscape, and use the source–sink approach. The average distance index of the source–sink landscape is used to describe the resilience of urban form.
(4)$$L_{\mathrm{ij}}=\frac{\sum_{i=1}^{n}\min c_{\mathrm{ij}}}{n},i=1,2,3\ldots ,m;\;j=1,2,3\ldots ,m,\;$$
(5)$$A_{m}=L/L_{\mathrm{ij}}$$
where L_ij_ is the average distance index of the source–sink landscape; c_ij_ represents the distance from an image i in the source to an image j in the nearest sink; n is the number of images in the source; and m is the number of images in the sink. The smaller the value of L_ij_, the better the source–sink morphology, the stronger the coupling, and the greater the resilience. Conversely, some source–sink areas may be spatially blocked, and when the source has a negative effect, it lacks the corresponding sink to counteract it and is less resilient. Am is the morphological resilience index and L is a constant. In this study, it was set as the mean value of 996.32 in the study area in 2000.

#### 2.3.5. Integrated Urban Resilience Index

Based on the Regional Multi-Dimensional Development Index (RMDI) [53,54], the comprehensive resilience index of the cities in the four southern regions was measured by the polygon method of RMDI. The polygon method is divided into two methods: sequentially aligned polygons and fully aligned polygons. The sequentially aligned polygon area method is to take a fixed point as the common point of the polygon area, obtained by calculating the area of the triangle formed by the adjacent line segments at the common point, and the polygon area is used as the value of the comprehensive index. In this study, based on the three-dimensional coordinates of scale, density and morphology, with O as the origin, OA, OB and OC represent the trigonal cone consisting of the scale resilience index, density resilience index and morphology resilience index, respectively (Figure 3). The volume of this triangular cone is defined as the integrated urban resilience index (N_A_), which is calculated as follows.
(6)$$N_{A}=\frac{1}{6}\left(A_{S}*A_{D}*A_{M}\right)$$
where A_S_ is the scale resilience index, A_D_ is the density resilience index and A_M_ is the morphological resilience index. The overall urban resilience index is the volume of the trigonometric cone produced by the combined effect of scale, density and morphology, and the greater the volume, the greater the overall urban resilience.

## 3. Results and Analysis

### 3.1. Scale Resilience

As can be seen from Figure 4, the scale resilience indices of the counties and cities in the study area varied considerably during the period from 2000 to 2020. Among them, the scale resilience index of all counties and cities in Aksu Administrative Offices, except for Xinhe County, showed a relatively small change, indicating that the construction land area of these counties and cities did not grow extensively during 2000–2020, and their urban development scope basically remained stable. However, the scale resilience index of Xinhe County showed an increasing trend during 2000–2015 and a decreasing trend after 2015. The scale resilience index of all counties and cities in Kizilsu Kirgiz Autonomous Prefecture remained basically unchanged during 2000–2015, but the index grew smaller in Akto County and Artux City during 2015–2020. The scale resilience index of all counties and cities in Kashgar Administrative Offices was more variable, among which the three counties and cities of Taxkorgan Tajik Autonomous County, Yecheng County and Bachu County were basically unchanged, but the scale resilience index of Zepu County, Kashgar City and Yengisar County showed positive growth after 2015. Hotan Administrative Offices had the lowest scale resilience index in the whole study area, and the counties and cities did not change much during the period of 2000–2015, but after 2015, the counties and cities of Moyu County, Pishan County, Minfeng County, Yutian County and Qira County showed a decreasing trend in their indices, while the indices of the other counties and cities showed an increasing trend. Overall, there was a wide variation in the scale resilience across the study area. For each state, the influence of population clustering in the central counties and cities of each state has made the area of construction land available for future construction smaller, resulting in the scale resilience index for non-central counties and cities in each prefecture being lower than the average for the region.

The scale resilience index for each district and county was divided into five categories: low resilience, medium-low resilience, medium resilience, medium-high resilience and high resilience, according to Table 4 and Figure 5. Figure 5 shows that between 2000 and 2020, the number of counties and cities with low resilience decreased from seven to five, with Akqi County and Hotan County increasing from low to medium-low resilience in 2020. The number of counties and cities with medium-low resilience increased from 6 to 10, with Zepu County increasing to medium resilience in 2010 and Kucha City, Kalpin County and Yutian County decreasing from medium to medium-low resilience. The number of counties and cities with a medium resilience level increased from 9 to 10, of which Aksu City and Moyu County decreased from medium-high resilience to medium resilience in 2020. The number of counties and cities with medium-high resilience increased from seven to eight and then decreased to six, among which Jiashi County was increased to medium-high resilience in 2005 and Yengisar County upgraded to medium-high resilience in 2010. There were three counties and cities with high resilience levels, indicating that these three counties and cities, Xin’he County, Awat County and Makit County, had larger scale resilience indices compared with the other counties and cities, indicating that they have a larger area of land available for urban construction in the future.

### 3.2. Density Resilience

Combined weights were used to evaluate the density resilience of the four southern regions of Xinjiang. The layer weights in the density resilience evaluation system were obtained by adding up the weights of the indicator layers, while the combined weights of the indicator layers were obtained by Equation (2). The weights of each indicator are shown in Table 5.

The density resilience indices of the 33 counties and cities in the four southern prefectures of Xinjiang in 2000, 2005, 2010, 2015 and 2020 were calculated by combining the weights. As can be seen from Figure 6, the density resilience indices of all counties and cities in the study area during 2000–2020 were all less than 0.2, indicating the low level of economic development in the study area. The density resilience index of Aksu Administrative Offices and Kashgar Administrative Offices was high, with the maximum value reaching 0.14 and 0.12, respectively, while the index values of Kizilsu Kirgiz Autonomous Prefecture and Hotan Administrative Offices were low, with the maximum value of Kizilsu Kirgiz Autonomous Prefecture being 0.025. This indicates that the economic development level of Aksu Administrative Offices and Kashgar Administrative Offices was much higher than that of Kizilsu Kirgiz Autonomous Prefecture and Hotan Administrative Offices. The counties and cities with the largest indices in Aksu Administrative Offices were Aksu City and Kucha City, and the largest index in Kashgar Administrative Offices was found for Kashgar, indicating that the economic development levels of these three counties and cities are among the highest levels in the study area. The level of resilience of the other counties and cities changed little between 2000 and 2020, indicating that the economic development of the region was slower and that it was more influenced by its geographical location.

According to Table 4, the density resilience indices of each district and county were classified into five categories: low resilience, low to medium resilience, medium to high resilience, and high resilience (Figure 7). It can be found that between 2000 and 2020, the number of counties and cities with a low resilience level changed from 5 to 4, among which Wucha County was upgraded to medium-low resilience level in 2010, indicating that the economic development level of this county and city improved during this period. The number of counties and cities with a medium-low resilience level increased from 6 to 8 after 2005 and then decreased to 4 in 2020. The number of counties and cities with a medium-high resilience level changed from 9 in 2000 to 7 in 2020. Aksu, Kuche and Kashgar always maintained a high resilience level, indicating that the economic development level and infrastructure construction of these three counties and cities belong to high level in the study area. Shache County changed to a high resilience level, while Yecheng County was upgraded from a medium-high to a high resilience level in 2015, and Moyu County directly changed from a medium to a high resilience level in 2020. As can be seen from Table 4, in the total weight ranking of the indicator layer, there are three indicators of economic resilience with weight values greater than 0.1, while there is only one indicator of social resilience with a weight value greater than 0.1, indicating that the economic and social development of the study area presents a large mismatch.

### 3.3. Morphological Resilience

CAS 30m land use data were taken as the data source, and the source and sink landscape classifications were obtained for 2000, 2005, 2010, 2015 and 2020, including gray landscapes (urban construction land), green landscapes (woodland and grassland), and blue landscapes (water bodies). Through the calculation of the average distance index of the source–sink landscape, a morphological resilience index was obtained for the study area.

As can be seen from Figure 8, the morphological resilience index varied considerably between 2000 and 2020. The small change in the morphological resilience index between 2000 and 2015 indicates that there was no significant change in the layout of the source–sink landscape between the counties, while in 2020, the morphological resilience index varied considerably among the counties, indicating that there was some change in the layout of the source–sink landscape among the counties. It is possible that the area of the gray landscape has reduced and the area occupied by the blue-green landscape has increased, thus allowing the counties and cities to adequately mitigate the negative impacts of the gray landscape and leading to an increase in their morphological resilience index.

The morphological resilience index of each district and county was classified into one of the five categories of low resilience, medium-low resilience, medium resilience, medium-high resilience and high resilience. According to the classification criteria in Table 4 (Figure 9), the results show that there were three counties and cities with low resilience in 2000 and only one with low resilience in 2020; nine counties and cities with medium-low resilience were found in 2000 and three in 2020. The number of counties and cities with medium resilience increased from 9 to 13, and the number of counties and cities with high resilience increased from 7 to 10. The number of counties and cities with high resilience in 2020 were mostly concentrated in Hotan Administrative Offices. Overall, the morphological resilience level of the study area increased between 2000 and 2020. Most of the morphological resilience levels are medium resilience levels and medium-low resilience levels.

### 3.4. Comprehensive Analysis of Scale-Density-Morphological Resilience

As can be seen from Figure 10, the urban resilience index of the counties and cities in the four southern prefectures of Xinjiang showed a general trend of first decreasing and then increasing from 2000 to 2020. It was found that the average values of the urban resilience index in the study area in 2000, 2005, 2010, 2015 and 2020 were 0.0210, 0.0207, 0.0150, 0.0163 and 0.0202, respectively. This indicates that with the development of urbanization, the 33 counties and cities in the four southern prefectures of Xinjiang have gradually strengthened their ability to withstand and adapt to disasters, and are able to achieve rational deployment of resources and recover quickly from disasters when they strike. By region, Aksu Administrative Offices was the county with the highest average urban resilience index in 2000, 2005, 2010, 2015 and 2020 (0.0370, 0.0318, 0.0255, 0.0254 and 0.084, respectively), with Kashgar Administrative Offices having the next highest average level of urban resilience, with averages of 0.0243, 0.0273, 0.0177, 0.0203 and 0.0196, respectively. In addition, the fluctuation in the curve of the urban resilience index in the past years showed that the study has been able to find that the urban resilience index was the most important factor in the development of the city. In addition, from the fluctuation in the curve of the urban resilience index in the past years, it can be seen that there were large differences in the high and low urban resilience of the counties and cities in the study area.

The comprehensive resilience indices of each district and county were divided into five categories: low resilience, medium-low resilience, medium resilience, medium-high resilience and high resilience, as shown in Figure 11. It can be seen that the comprehensive resilience levels in the study area varied widely between 2000 and 2020, with Aksu City and Kashgar City consistently maintaining high resilience levels, which indicates that Aksu and Kashgar City have been improving in terms environmental quality while carrying out socioeconomic development within a certain scope of urban construction during these 20 years. Before 2010, the comprehensive resilience levels of the counties and cities were mostly low resilience levels, but after 2010, these gradually changed to medium resilience levels, but the number of counties and cities with medium-high and high resilience was very small. This indicates that the counties and cities were able to respond better to external influences after 2010, but due to their geographical location, the urban development was slower.

## 4. Discussion

Urban resilience is of great importance in urban master planning. To ensure the safe development of the city, in addition to building an urban disaster defense system to adequately cope with the situation, we should also pay attention to the practical significance of urban scale, density and form. On this basis, scientific and reasonable planning should be carried out.

The difference between the scale and density resilience indices and the average level of the study area has been decreasing, and the overall trend is deteriorating. Compared with more developed areas such as the eastern seaboard, the study area shows a worse situation in terms of its resilience portfolio, and the trend of improvement is weaker over the years. The difference in resilience between districts and counties is much greater than in the east, and the development is highly uneven. Due to the slower development process in this region than in the east, the characteristics of the portfolio have not been transformed and different cities are facing the problem of reduced resilience and lack of security under the urbanization process.

Urban resilience is not only an extension of the concept of sustainable development, but also includes, to some extent, related concepts such as ecological protection and harmony between people and places [55,56]. As an important initiative to build resilient cities, most of the research objects have focused on areas with good economic development levels, while the construction of resilient cities in poor border areas has not been addressed. Therefore, this paper takes the poor border areas as the research object [57,58,59,60] and evaluates the urban resilience of such areas in terms of both ecological protection and economic development. However, due to the specificity of the location of the study area, it is difficult to obtain data, so the assessment of density resilience is not comprehensive.

## 5. Conclusions and Recommendations

### 5.1. Conclusions

Since the study area contains desert and other unavailable land, its available urban construction area is small, and the urban development faces a strong constraint of scale safety. Compared with the average level of the study area, the scale resilience of counties and cities in Aksu and Kashgar Administrative Offices were higher than the average level, while most of the counties and cities in Hotan and Kashgar regions were lower than the average level; the differences among the counties and cities are mainly due to the uneven development of land urbanization.

The study area is a poor border area with less developed ideology, production methods and technology, which seriously restricts the development of local society and economy. The density resilience of the counties and cities in the study area varies greatly, and the density resilience of Aksu, Kashgar and Kuche is much higher than that of other counties and cities. The morphological resilience indices of the counties and cities in the study area varied considerably during the study period, but the overall trend was upward, and the landscape pattern changed significantly. This indicates that the ecological protection of the study area has been effective.

The combination of scale-density-form resilience can better reflect the safety level of cities. In the past 20 years, the study area has not been transformed into a resilient combination of counties and cities, although it is relatively stable.

### 5.2. Scale Resilience Regulation Path

In order to control urban sprawl, attention should be paid to the increase in urban built-up areas in urban planning, which in turn should guide the decision making and reasonably delineate the town development boundary according to the maximum degree of urban expansion. Due to the limitations of topography and landscape, there is less space for the development of the main urban areas in the counties and cities of the four southern Xinjiang prefectures, therefore it is necessary to avoid excessive concentrations of construction to form landscape pressure, but in order to achieve a benign and balanced development between ecology and economy, the following should be carried out:

Areas with high building density such as Kashgar City, Aksu City and Kucha City should reasonably limit development intensity and avoid concentrated construction, but the stock of land and low-efficient use of land in other areas should be moderately developed and the attention and utilization rate of abandoned land should be improved. In addition, the renovation and renewal of urban villages should be strengthened and the intensity of landscape construction should be balanced.

The balanced development of urban and rural areas should be promoted, including arranging industrial layout and development resources in an integrated manner, coordinating urban and rural environmental protection and remediation as well as economic development, avoiding differences in resilience caused by excessive urban–rural disparities, strengthening the role of urban areas in radiating and driving the county, and improving the overall scale resilience.

### 5.3. Density Resilience Modulation Pathway

In order to enhance the density resilience, the industrial structure should be optimized, the intensive use of resources should be improved, and the quality of the population should be improved, specifically to:

Optimize the industrial structure; promote the development of secondary and tertiary industries; pay attention to the adjustment and transformation of high energy-consuming industries to avoid waste of resources; and optimize and upgrade the energy production–consumption structure to reduce the excessive consumption of the ecological environment.

Investment in education should be increased to optimize the demographic structure and improve the quality of the population. Higher population quality is conducive to alleviating the relationship between the population and resources and the environment, reducing the consumption of resources, and showing higher subjective initiative when facing urban security issues, actively adapting to the uncertainty issues in urban development, which is conducive to enhancing urban resilience.

### 5.4. Morphological Resilience Regulation Path

It is crucial to protect woodland resources and characteristic landscape resources, maximize the ecological role of woodlands and other green patches, protect and optimize green corridors, leave channels for biological migration and build activity networks, in addition to improving the integrity and systemic nature of green spaces and providing ecological space for urban development.

Moreover, it is important to make good use of the blue-green natural landscape and combine it well with urban construction land into several functional groups, form an excellent urban ecological pattern, strictly protect the group isolation green belt, prevent the disorderly spread of urban construction land, and enhance morphological resilience.

## Figures and Tables

**Figure 1 ijerph-20-05106-f001:**
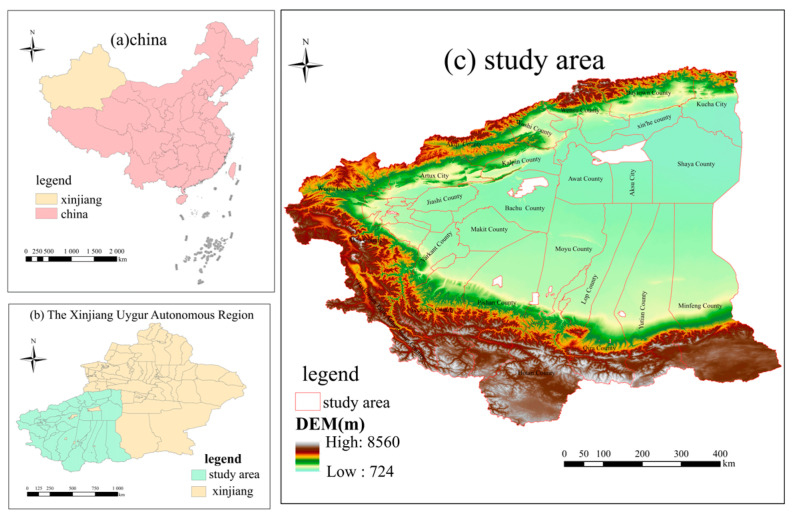
(**a**) Location of Xinjiang Uygur Autonomous Region in China; (**b**) location of the 33 counties and cities in the four prefectures of southern Xinjiang; (**c**) study area.

**Figure 2 ijerph-20-05106-f002:**
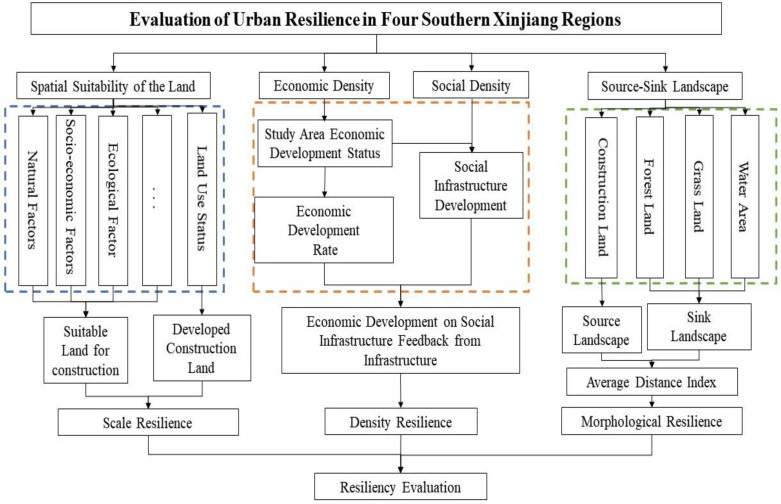
“Scale-density-form” urban resilience evaluation technology roadmap.

**Figure 3 ijerph-20-05106-f003:**
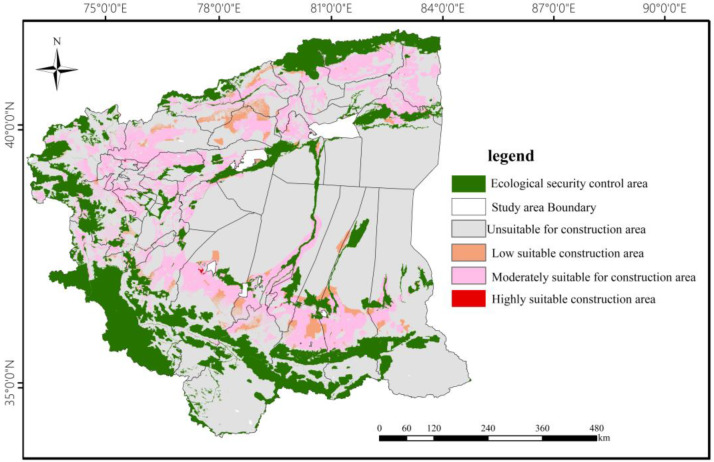
Comprehensive map of construction land suitability evaluation in the four southern Xinjiang regions.

**Figure 4 ijerph-20-05106-f004:**
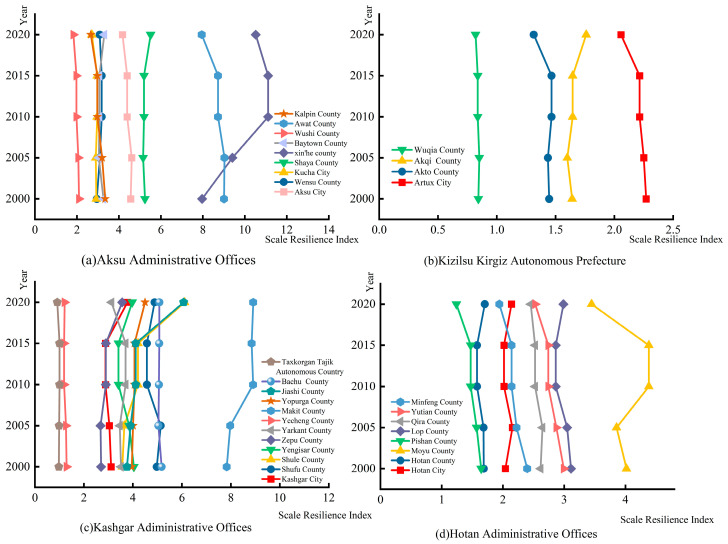
(**a**) Scale resilience index of Aksu Administrative Offices from 2000 to 2020. (**b**) Scale resilience index of Kizilsu Kirgiz Autonomous Prefecture from 2000 to 2020. (**c**) Scale resilience index of Kashgar Administrative Offices from 2000 to 2020. (**d**) Scale resilience index of Hotan Administrative Offices from 2000 to 2020.

**Figure 5 ijerph-20-05106-f005:**
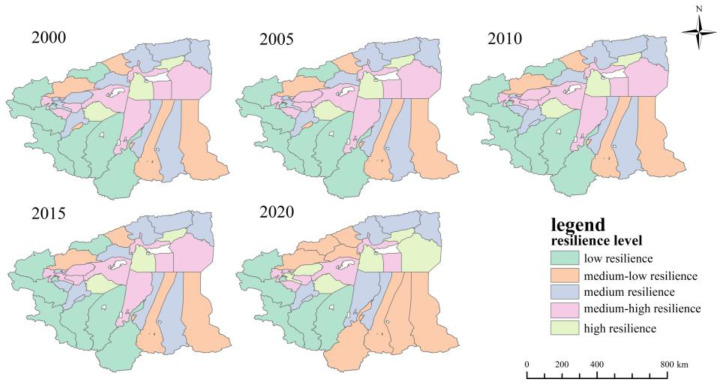
Scale resilience level distribution of four prefectures in southern Xinjiang from 2000 to 2020.

**Figure 6 ijerph-20-05106-f006:**
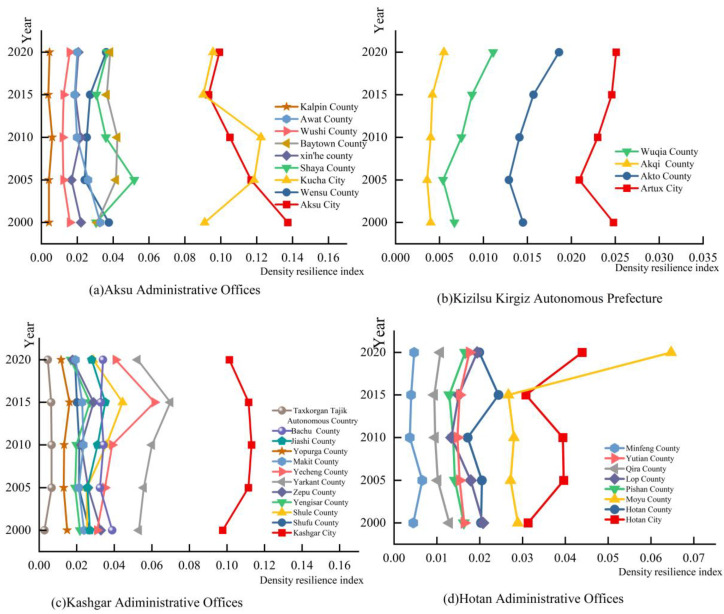
(**a**) Density resilience index of Aksu Administrative Offices from 2000 to 2020. (**b**) Density resilience index of Kizilsu Kirgiz Autonomous Prefecture from 2000 to 2020. (**c**) Density resilience index of Kashgar Administrative Offices from 2000 to 2020. (**d**) Density resilience index of Hotan Administrative Offices from 2000 to 2020.

**Figure 7 ijerph-20-05106-f007:**
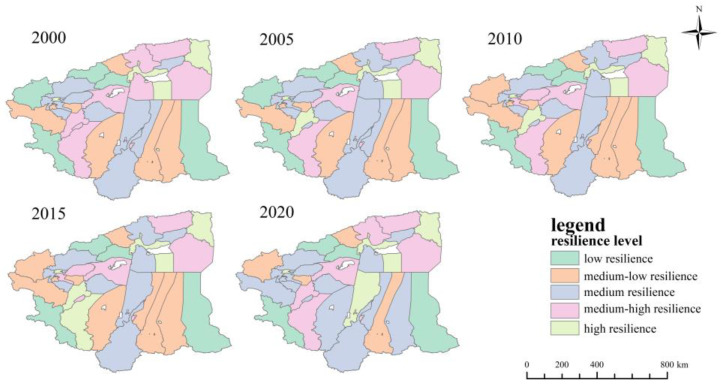
Distribution of density resilience level in four prefectures of southern Xinjiang from 2000 to 2020.

**Figure 8 ijerph-20-05106-f008:**
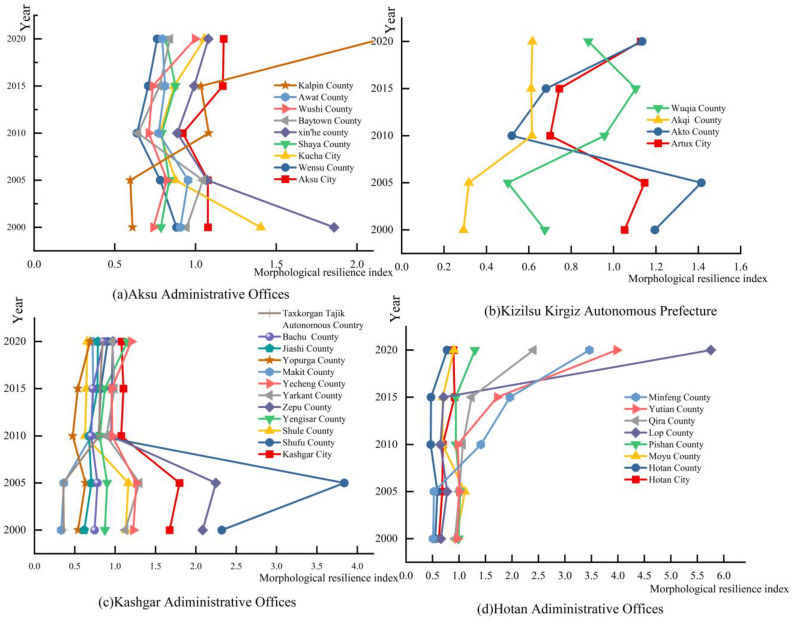
(**a**) Morphological resilience index of Aksu Administrative Offices from 2000 to 2020. (**b**) Morphological resilience index of Kizilsu Kirgiz Autonomous Prefecture from 2000 to 2020. (**c**) Morphological resilience index of Kashgar Administrative Offices from 2000 to 2020. (**d**) Morphological resilience index of Hotan Administrative Offices from 2000 to 2020.

**Figure 9 ijerph-20-05106-f009:**
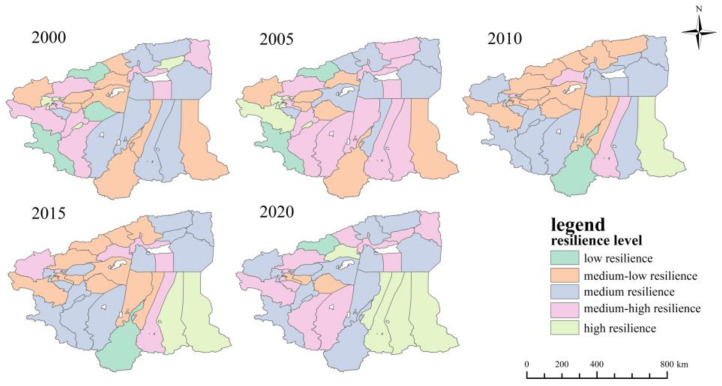
Distribution of morphological resilience level of four prefectures in southern Xinjiang from 2000 to 2020.

**Figure 10 ijerph-20-05106-f010:**
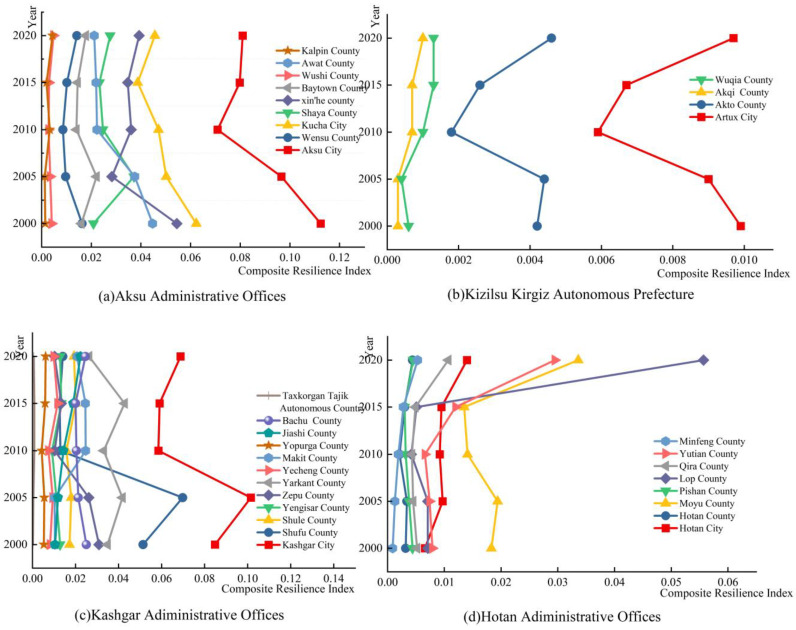
(**a**) Comprehensive resilience index of Aksu Administrative Offices from 2000 to 2020. (**b**) Comprehensive resilience index of Kizilsu Kirgiz Autonomous Prefecture from 2000 to 2020. (**c**) Comprehensive resilience index of Kashgar Administrative Offices from 2000 to 2020. (**d**) Comprehensive resilience index of Hotan Administrative Offices from 2000 to 2020.

**Figure 11 ijerph-20-05106-f011:**
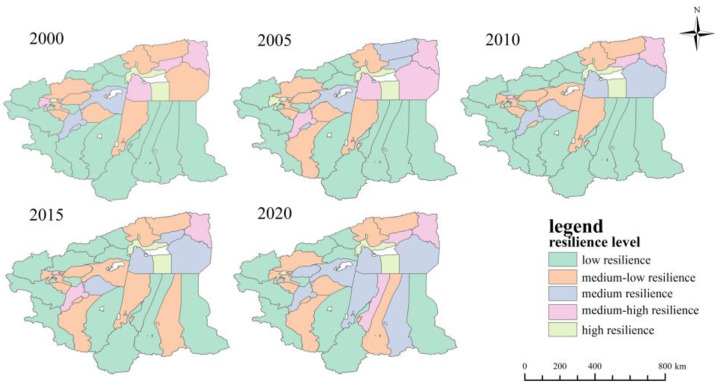
Distribution of comprehensive resilience level of four prefectures in southern Xinjiang from 2000 to 2020.

**Table 1 ijerph-20-05106-t001:** Data required and their sources.

Data Categories	Required Data	Source	Years
Spatial Data	Digital Elevation Model	Geospatial data clouds (http://gscloud.cn) format: 23 March 2022	2020
	Vegetation cover index data	NASA (https://ladsweb.nascom.nasa.gov/) format: 23 March 2022	2000/2010/2020
	Population density	World Population Dataset (https://www.worldpop.org) format: 25 March 2022	2000/2010/2020
	Road network data	National road vector data to2020 (https://www.openstreetmap.org/) format: 26 March 2022	2020
	Land use data	CAS 30 m land (https://www.resdc.cn) format: 18 March 2022	2000/2010/2020
	Ecological Protection Red Line	Courtesy of the subject team	2020
Socioeconomic Data	Total population at end of year	Xinjiang Statistical Yearbook County Statistical Yearbooks (https://navi.cnki.net/) format: 2 April 2022	2000/2005/2010/2015/2020
	Number of beds in hospitals and health centers		
	Number of students in general secondary schools		
	Number of pupils in primary schools		
	Gross Domestic Product		
	Value added of secondary industry		
	Total retail sales of social consumer goods		
	Fiscal revenue		
	Total output value of agriculture, forestry, animal husbandry and fishery		

**Table 2 ijerph-20-05106-t002:** Suitability evaluation index system of construction land.

Assessment Factors	Evaluation Factors	Grading Criteria			
		Appropriate Height (4)	Moderate Suitability (3)	Low Suitability (2)	Unsuitable (1)
Natural conditions	Slope	<5°	5°–15°	15°–25°	≥25°
	Elevation	<1757	1757–3012	3012–4427	≥4427
	Vegetation Cover Index	<0.0774	0.0774–0.1408	0.1408–0.2309	≥0.2309
Socioeconomic factors	Population density	≥150	60–150	15–60	<15
	Distance from main traffic routes	<500	500–1000	1000–1500	1500–2000
	Type of land use	Building Sites	Arable Land	Woodland, Grassland	Other
Ecological factors	Ecological protection red line	Outside the ecological protection red line			Within the ecological protection red line

**Table 3 ijerph-20-05106-t003:** Evaluation index system of urban density.

Assessment Factors	Evaluation Factors	Specific Indicators
Density resilience	Social density	Total population at end of year
		Rural practitioners
		Number of beds in hospitals and health centers
		Number of students in general secondary schools
		Number of pupils in primary schools
	Economic density	Gross domestic product
		Value added by secondary industry
		Total retail sales of social consumer goods
		Fiscal revenue
		Total output value of agriculture, forestry, animal husbandry and fishery

**Table 4 ijerph-20-05106-t004:** Classification criteria of resilience level in the study area.

Type	Low Resilience	Medium-Low Resilience	Medium Resilience	Medium-High Resilience	High Resilience
Scale Resilience	<1.6875	1.6875–2.7028	2.7028–3.7560	3.7560–5.2441	>5.2441
Density Resilience	<0.0067	0.0067–0.0163	0.0163–0.2890	0.2890–0.0530	>0.0530
Morphological Resilience	<0.3641	0.3641–0.7455	0.7455–0.9942	0.9942–1.4044	>1.4044
Combined Resilience	<0.0082	0.0082–0.0209	0.0209–0.0349	0.0349–0.0623	>0.0623

**Table 5 ijerph-20-05106-t005:** Weight of density resilience index determined by different methods.

Evaluation Factors		Specific Indicators		
Indicators	Weights (Combined Method)	Weighting		
		Entropy Method	Hierarchical Analysis Method	Portfolio Weights
Social Density	0.2968	0.0512	0.2143	0.1092
		0.0575	0.0714	0.0409
		0.0840	0.0714	0.0597
		0.0610	0.0714	0.0434
		0.0614	0.0714	0.0436
Economic Density	0.7032	0.1072	0.0665	0.0710
		0.1509	0.1195	0.1796
		0.1775	0.1195	0.2112
		0.0637	0.0973	0.0617
		0.1857	0.0973	0.1799

## Data Availability

Not applicable.

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
