# Peer review of "Analysis of the Spatial and Temporal Evolution of Urban Resilience in Four Southern Regions of Xinjiang"

_ijerph, 2023, doi:10.3390/ijerph20065106_

Round 1

Reviewer 1 Report

1)    The last sentence of the second paragraph on page 2 is confusing, “urban resilience has been the focus of urban resilience research”, revision is needed. 

2)    The literature review did not show how urban resilience was innovatively defined in this study, i.e., scale, density, and form, although it was said that scholars in China took this. More elaboration is needed to validate the scale_density_form framework. The last paragraph of Section 2.3.1 is not very convincing why these three aspects are important, i.e., need references. 

3)    Figure 1 was referred to in the text.

4)    Table 1 should list the year of the data, given the fact that some data may not be available every year. If it is the case, how was the missing year data treated?

5)    A summary of the data will be needed, not just saying where they were obtained from. 

6)    Shall we use scale, density, form instead of scale_density_form? 

7)    The paragraph of Section 2.3.2 did not lead to the content presented in Table 2. Need revision. 

8)    For the grading criteria in Table 2, what does it mean by appropriate “height”? Say, Is population density of  >150 the appropriate height?  Very confusing. 

9)    The term economic resilience and social resilience must be described, of course, in reference to the literature 42 to 44. 

10)                  Table 3 introduced a new term “layer”. This was not consistent with the other two dimensions in Table 2 and Table 4. 

11)                  In the paragraph before equation (1) and (2) said the entropy value method was used, so further explanation is needed here. 

12)                  The results presented in Table 5 needed a proper explanation of the methods to be described in Section 2.3.3. That is, what is the hierarchical analysis method ? How the results were obtained. So does the entropy value method. 

13)                  Figure 2, 4, 6, and 8 should be enlarged and improved in quality (in terms of resolution or sharpness). The axes were strange. There was no axis title. 

14)                  Explanations of source-sink in Section 2.3.4 were, again, very confusing and not informative.

15)                  Explanation of equation 5 was not understandable. How were As, Ad, Am obtained? Why was it divided by 6? Adequate reference in addition to [48] is needed. 

16)                  Why was equation 6 described in Section 3.1, rather than Section 2.3.2.  

17)                  The reading of the results, after Figure 2, 4, 6, etc. were very locally specific, so not very useful for the (international) general audience. 

18)                  Table 5 had portfolio weights but it was not described before. 

19)                  Check capitalization of the title of Table 2, 3, 4, and Figure 9. 

20)                  The discussion and conclusion in Section 4 and 5 were superficial, and sound more or less a summary. 

21)                  It is advised to clearly state the contribution of the study with strong evidence of the results and findings presented in the paper. 

Author Response

-

Reviewer 2 Report

The article, entitled “Analysis of the spatial and temporal evolution of urban resilience in four southern regions of Xinjiang”, presents a case study method of analysis with the development of a three-dimensional resilience analysis framework based on scale, density and form. The manuscript has the potential to contribute relevant new insights on urban resilience and governance planning.

Regrettably, the manuscript is not formatted to include the line pagination. Therefore, I will not attempt to detail some of the errors I located. There are several typo and grammatical errors in the introduction section as well as the other sections. The paper should be edited professionally to correct these issues. Some of the paragraphs are awkwardly written and should be broken into smaller sentences to discuss the points more clearly.

What is missing from my reading of the urban resilience literature is how you structure your three indicators from your framework. You discuss how urbanization can also pose as a maladaptive practice. However, the framework seemingly appears to embrace urbanization as a positive factor and encourages this as a positive aspect for urban resilience governance. In other parts of the paper it appears to contradict this position and it is not clear what aspects of resilient “scale, density and form” are desired for each research site. This does not account for the nuances in the resilience debates that are well debated such as, community resilience, household resilience, economic resilience, disaster resilience, etc. Each of these resilience indicators define different attributes and indicators to measure resilience. This part of the discussion seems to be overlooked in your literature review as well as methodological approach. How does this study (framework) account for these differences and measurements? Is it not possible to have different resilience terms or measurements from one region, one city or province to another? Would not one region determine one attribute to meet resilience and another disagree? This would also speak to the economic variation of the regions. Some areas of the study site seem to imply they have more rural attributes rather than urban centers. If the framework targets urban resilience, then logically these would not be appropriate site locations for the framework to measure. Why or why not are these included in the study assessment?

There seemed to lack information for a better rationale as to why the study site location was chosen. The concluding remarks appear to take a rather pro-economic and political stance as to why the region is encouraged to undertake urbanization processes. This is perhaps a bit misleading and can be addressed in part by providing better argumentation for the why the area needs to be studied, the differences between the urban spaces that will be examined or excluded (is the focus solely on urban spaces?) and how the study aims to contribute to the missing body of literature.

The framework indicators selected also appear to be missing the links to the literature. How does the current body of literature define these indicators, how are they measure and how do they inform our analysis and understanding of urban resilience? The density indicator seemingly purports that urbanization is indeed a positive correlation to urban resilience. Perhaps this might be the case, but I am rather doubtful given the enormous range of resilience literature publications that discusses the nuances between governance regimes and communities that impact differences and vulnerabilities. The framework excludes these differences yet, praises these attributes in the conclusion as to why economic development is an appropriate solution and this comes across as a very political stance. Rather the framework and results should speak as to why it is appropriate to use this method to measure urban resilience versus other approaches currently in practices. One questions that should be answered is what is the potential of applying remote sensing and GIS to urban resilience measurements? Why is the urban resilience index the most important factor in the development of a city? How do you account for the peri-urban and rural regions that tied to urban-rural economies?

Lastly, the abstract could be revised to better reflect the contributions of the three-dimensional framework and how it answers important questions centered on defining and measuring urban resilience.

Author Response

-

Reviewer 3 Report

The tasks of analyzing and evaluating the urban agglomeration in
regional development on the principles of sustainable development are
very important and interesting. However, some questions are not
clearly described in the article:

In paragraph 2.3.3, two levels are proposed for assessing the
sustainability of a territory in terms of density: social and economic
sustainability. It is not very clear why only two levels have been
chosen and why such important levels for density assessment as, for
example, transport or environmental are not considered?
Also, in paragraph 2.3.3, it is proposed to use the methods of Wu Jiaqi,
Li Jintao, Yao Zhen to assess the sustainability of cities for research.
But the article does not provide a justification for the choice of
these methods and their advantages. Similarly, paragraph 2.3.5 refers to
the use of the method
polyhedron proposed by Xu Yong to measure the complex index the
sustainability of each county and city in the study area, but also
without substantiating the advantages of the chosen method.
To determine the importance weights for each indicator, the authors used
the entropy value method and the hierarchical analysis method, and then,
as the authors write, these two methods were combined to obtain the
final weights. I would like a more detailed explanation of exactly how
these methods were combined.
In general, the article is logically structured and useful for
researchers working in this field. If the questions raised are
clarified, the article can be recommended for publication.

Author Response

-

Round 2

Reviewer 1 Report

The manuscript has been much improved than the previous version. However, if there is a figure showing the flow of the study, reading will be much easier. In addition, recheck pagination. 

Reviewer 2 Report

Thank you for your revisions and detailed responses.

Although the authors provide a short rationale for their methodological approach (RS/GIS), this information seems to be missing in the manuscript. This seems to be an important contribution the authors are making but is seemingly omitted from their supporting argument in the introduction to provide evidence for the analysis, methods design, results and overall contribution to the resilience discipline.
